# Ovarian Stem Cells for Women’s Infertility: State of the Art

**DOI:** 10.3390/biomedicines12061139

**Published:** 2024-05-21

**Authors:** Krzysztof Grettka, Katarzyna Idzik, Katarzyna Lewandowska, Ksena Świętek, Simone Palini, Franco Silvestris

**Affiliations:** 1FERTILITA Sp. z o.o., 41-700 Ruda Śląska, Poland; 2GLOBIANA Sp. z o.o., 40-083 Katowice, Poland; 3Cervesi Hospital, AUSL Romagna, Via Beetovhen 1, 47841 Cattolica, Italy; 4OSC Research Project, University of Bari Aldo Moro, 70121 Bari, Italy

**Keywords:** cancer treatment-related infertility (CTRI), infertility, oocyte-like cells (OLCs), ovarian stem cells (OSCs), premature ovarian failure (POF), stem cells

## Abstract

Today, women’s infertility is considered a social disease in females, occurring not only as an effect of POF (premature ovarian failure) but also as CTRI (cancer treatment-related infertility) in oncologic patients. Several procedures for FP (fertility preservation) are currently adopted to prevent this condition, mostly based on utilization of retrieved eggs from the patients with subsequent IVF (in vitro fertilization) or cryopreservation. However, great interest has recently been devoted to OSCs (ovarian stem cells), whose isolation from female ovaries, followed by their in vitro culture, led to their maturation to OLCs (oocyte-like cells), namely, neo-oocytes comparable to viable eggs suitable for IVF. Translation of these data to FP clinical application creates new hope in the treatment of infertility. Thus, in line with the significant progress in using stem cells in the regenerative medicine field, neo-oogenesis via OSCs, which is currently unapplicable in fertility preservation procedures, will provide novel possibilities for young and adult females in motherhood programs in the future.

## 1. Introduction

Infertility in women with premature ovary failure (POF) and in females treated with anti-cancer drugs, namely cancer treatment-related infertility (CTRI), leads to a permanent disease that precludes programs for a normal reproductive life. For its diffusion worldwide, infertility appears as a nearly social disease in most Western countries, often as an effect of intense styles of life and cancer diffusion, as well as in the third world due to the higher incidence of malnutrition and infectious diseases other than cancer [1].

Several procedures are currently adopted in fertility preservation (FP) programs. The most common one is based on the retrieval of eggs picked up from ovaries after their maturation via hormone stimulation, followed by prompt in vitro fertilization (IVF) through sperm injection. This method is largely used in women with POF, whereas in women with cancer, the recruited eggs are cryopreserved with consent for IVF after healing from cancer. However, this procedure is not favorably recommended because of its intrinsic oncogenic risk, especially in young females bearing hormone-sensitive cancers, such as the majority of breast and gynecological tumors. On the other hand, egg maturation requires three weeks following the hormonal burst, and this time delay is improper when anti-tumor therapies are urgently needed.

A procedure suitable for patients with hormone-sensitive tumors, particularly in pre-puberal age, is provided in ovarian cortex cryopreservation and reimplantation after cancer. This hormone-free method is based on freezing and subsequent replacing of native ovarian fragments and appears to be a suitable choice for pre-puberal or young patients, though its successful pregnancy rate is lower by as much as one-third compared to egg pick-up, whose average outcome rate was assessed from 45% of IVF procedures. Besides the low results in terms of live births, namely, less than 15%, it has indeed been reported that cortical reimplantation also implies a risk of regenerating the original cancer, since tumor cells could be present in the cryopreserved tissue [2].

Another extreme FP approach to consensual maternity in infertile females is ovo-donation from healthy women, with intrauterine implantation of embryos and surrogative pregnancy. However, this needs to be regulated by specific rules in relation to certain ethical concerns.

Recently, the field of regenerative medicine has been favorably implemented by using stem cells in a variety of human applications, and the discovery of ovarian stem cells (OSCs) in women provides novel potential for future utilization in FP programs [3]. In fact, based on the possibility of expanding oocyte-like cells (OLCs) from OSCs to obtain mature oocyte populations, it appears possible to select viable OLCs suitable for FP programs to treat both POF and CTRI with no oncogenic risk for patients and without restrictions related to the timing of oocyte maturation [4]. Furthermore, the availability of selected populations of OLCs from OSCs also provides the opportunity to freeze consistent numbers of OLCs to provide a personal repertoire for each patient, which could be enough for IVF after cancer.

Here, we revise the major knowledge and the state of art in OSC research for future application in women’s FP by reviewing data from the medical literature, as well as from our experimental studies from a scientific project supported by the European Union.

## 2. Discovery of Ovarian Stem Cells

Despite the central dogma that oocyte progenitors are fixed at birth in mammalian females and that menopause reflects the exhaustion of such a reserve in the ovaries, original studies from basic scientists in 2004 demonstrated the presence of stem cells in the postnatal ovarian cortexes of mice, namely OSCs, which were hypothetically capable of replenishing the ovarian reserve in relation to oocyte depletion. By using animal models of sterilized female mice, they proved that the transplantation of an ovarian cortex including OSCs resulted in neo-oogenesis and regeneration of fertility, thus implying that in the adult life of mammalian females, OSCs are presumably committed to supplementing the consumption of the oocyte reserve in the ovaries [5].

This discovery elicited enthusiasm in several groups of investigators, since neo-oogenesis via OSCs was confirmed in other mammalian animals, including rats, goats, and pigs, and these cells were also characterized by specific membrane markers, which were then useful for their isolation from the ovarian cortex in subsequent studies devoted to defining their cellular and molecular aspects, as well as their functional properties. Once again, these studies provided new evidence that the old paradigm of an ovarian reserve based on fixed numbers of oocyte progenitors needed to be revised in relation to confirmed neo-oogenesis via OSCs.

Studies in women have also demonstrated that OSCs are detectable within the ovarian cortex in both the pre- and post-menopausal ages (Figure 1).

By using isolation methods binding specific molecules to the OSC membrane, it has been possible to purify these cells from ovary cortical fragments and further investigate their functional aspects using in vitro cultures. Results from these studies prove that women’s OSCs exhibit smaller sizes than mature oocytes, usually lower than 10 μm, and that the consistency of this cell population within the female ovarian cortex is quite little compared to other cells, approximately 5%. In addition, this magnitude usually declines post-menopause. However, once isolated, using proper culturing conditions, OSCs are maintainable as viable cells in vitro. During a three-week extended culture, they gradually enlarge their size and acquire both morphology and measurements typical of cells bearing oogonial lineage—namely, round-shaped cells with a large nucleus and a diameter of 50–70 microns or more [6].

Recently, novel technologies for basic research in female OSCs have improved their isolation from cultures by using cell sorters capable of capturing a single OSC to better investigate the molecular pattern of these cells for translational application in FP programs. In fact, besides the possibility of obtaining single cells for culturing procedures, molecular assessment showed that when matured to oocyte-like cells (OLCs) in vitro, the original pattern of stem cells appeared to be modified for both a lack of membrane molecules typical of oogonial cells and the expression of other molecular biomarkers typical of mature oocytes. Moreover, when investigating the chromosomal content in OLCs early after their three-week culture period, it was proved via proper scientific methodology that they contained only one half of the chromosomal repertoire [7]. This biological condition is named ‘haploidy’, i.e., the halved content of the oocyte chromosomes to be reconstituted in ‘diploid’ shape after fusion with sperm. Based on these interesting peculiarities of OLCs derived from OSCs, as well as their functional properties, several investigators have expressed great curiosity and interest in the potential translation and clinical application of OSCs to subsequent FP programs for women with POF and CTRI.

However, despite the potential of OLCs for regeneration of oogenesis in infertile women, several investigators have expressed a few skepticisms in OSC capability to generate oocytes suitable for IVF [8]. The main concern is related to their low number within the female ovarian cortex, as well as the unknown physiological role of the OSCs in post-menopausal age. While the scarcity of the cells can be easily overcome through in vitro culturing and enrichment of expanded populations, their role in women’s advanced age is thought to be functional, recovering the ovary endocrine functions commonly deranged in the menopausal state. Such a potential role of OSCs in aged women inspires further interest from researchers and clinical investigators in this fascinating topic in molecular medicine, and novel studies are undoubtedly needed to provide more knowledge before their promising translation to the FP field. In reality, based on their suitability for being isolated and grown in cultures, there is a great expectance in infertility treatment programs, mainly in young women with cancer, to induce neo-oogenesis via OSCs rather than forcing oogenesis via a hormonal burst that, as described, provides additional oncogenic risk.

During the current precision medicine era, using OSC-derived OLCs would open novel avenues in FP procedures, which could utilize these cells for the usual IVF procedures or even for implantation in the ovaries of patients with POF or CTRI. Moreover, in CTRI patients, the availability of expanded populations of OLCs would offer the possibility of preserving personal repertoires of these cells to be repeatedly used in future after healing from cancer. 

## 3. Recruitment and Growing of OSCs

An important discovery in the past few years promoting relevant advancement in OSC research studies concerns the identification of special proteins expressed by these cells. In this regard, in addition to other molecules related to their stemness state, OSCs express several molecules as Ddx4 and Fragile on their surface, namely the functional receptors of these cells, which are typical of the oogonial lineage and are exposed only during the earliest phase of their development. Therefore, by applying dedicated reagents to these OSC receptors, it has been possible to isolate the cells and investigate their biological properties. The reagents commonly used against both Ddx4 and Fragile molecules include specific antibodies which can be easily linked to cell-revealing chemicals, such as fluorochromes, or solid supports useful for separating the OSCs. A support linking those antibodies is offered by magnetic microbeads, allowing the immunomagnetic absorption of those cells and their specific isolation.

The technical procedure to isolate OSCs from the ovarian cortex is thus based on recruitment of a small biopsy of this structure and subsequent processing of the tissue to isolate and culture OSCs. Briefly, the ovarian cortex fragment obtained through laparoscopy is quickly processed by collagenase to dissolve the cellular component and generate a cellular suspension including all cells present in the original piece. Then, this cell population is incubated with antibodies that capture the OSCs and subsequently passed through a magnetic field to discharge the cells that are not bound to both anti-Ddx4 and anti-Fragile antibodies conjugated to the magnetic beads [9]. Such a methodology of magnetic immunoselection is highly efficient in separating specific cell populations expressing unique markers, such as OSCs, from other components of relative cellular components.

Once the OSCs have been isolated, a detailed evaluation verifying both the purity and magnitude of this population is necessary. Flow cytometry analysis using fluorochromes to detect OSCs linked to anti-Ddx4 and anti-Fragile antibodies provides excellent responses for this investigation (Figure 2).

After repeated washes of the OSC suspension, the cells are then plated in dedicated wells and microchambers and incubated at 37 °C in CO_2_ atmosphere, namely in sterile incubators. Before plating the OSCs in culture, to favor continuity with the in vivo conditions, it is preferable to prepare a feeder layer of other cells to recreate a bed on which they will easily maintain viability and enlarge their size. To this end, the adoption of other cells from the patient providing the ovarian cortex sample appears quite functional. The mesenchymal stem cells (MSCs) from subcutaneous fat or from peripheral blood are suitable to resemble the in vivo conditions, promoting both steady maintenance and maturation of OSCs in vitro. However, a critical point to ensure the best establishment, growth, and maturation of OSCs in vitro is related to the culture medium. In fact, several components, such as proteins, female hormones such as FSH (follicular stimulating hormone), estradiol, and LH (luteal hormone), and growth factors, including EGF (epidermal growth factor), need to be added to the culture at different concentrations in relation to the progression of maturation of OSCs in culture. The full procedure for isolation and culture of OSCs is briefly depicted in Figure 3.

The microchambers including the OSCs also need to be repeatedly inspected during the culture to evaluate their size enlargement. With regards to this, OSCs undergo a progressive increase in their diameter. Since the original size is equal to 7–10 μm of diameter upon isolation, in the first week of culture, they grow up to approximately 20 μm and gradually increase during the next weeks. After the third one, their size is enlarged to reach about 50–70 μm of cell diameter, a size near that of mature oocytes, which are usually around 80–100 μm. Moreover, during their maturation, the OSCs undergo further morphologic changes, including condensation of both cytoplasmatic and nuclear components, and in the last days of culture, they extrude a portion of the nuclear content through a surface vesicle termed the ‘polar body’, which includes the full chromosome repertory, each in half form. Based on their size and morphologic variations, OSCs have undoubtedly become OLCs by this stage of maturation, and their polar body extrusion is physiologically related to the potentiality to be fertilized by sperm. In fact, when oocytes reach this differentiation as ‘haploid’ cells, they are classified in MII (metaphase II), since they are ready to reconstitute the full gene repertoire after fusing their half-chromosomes with the correspondent ones from sperm, which is possible only by suppressing a half part of the oocyte gene stock. Figure 4 depicts the progressive enlargement of OSCs from the original isolation to their maturation to OLCs, providing the possibility of using them for IVF. Gene analysis of OLCs from previous studies has shown that OLCs express the genes of mature oocytes; thus, they would possibly be suitable to be injected for fertilization with the male gamete.

Typically, both maturation of OLCs from OSCs and the number of mature oocytes obtained are a function of the original content of the samples processed. This is due to the low concentration of OSCs in the ovarian cortex, whose percent rate is estimated, as mentioned, to be about 5% with respect to the full cell population. However, based on the Ddx4^+^-cell population recovered from each sample, it is reported that a percentage equal to approximately 10% undergo their final differentiation to OLCs and potential oocytes.

## 4. Recent Results and Perspectives

In a recent clinical study, we investigated the feasibility of obtaining OLCs from a number of females prevalently affected by primary infertility as POF, as well as from a single patient with CTRI. This patient group was enrolled with the approval of the Health Silesian Authorities and in agreement with their regulations, and all patients provided their informed consent before donating their ovarian biopsies for research purposes.

The samples were processed in line with previous scientific work, and major results are reported in Table 1.

As shown, infertility with no pregnancies in all patients was prevalently due to POF, whereas a unique patient with CTRI was included in the study. Their ages ranged from 30 to 44 yrs, with a mean age of about 36 yrs. As expected, low serum levels of anti-Müllerian hormone (AMH) were found in all subjects with a minimal value equal to 0.01 ng/mL in most of them, including the patient with CTRI. The mean value of AMH measured from our cohort was calculated to be much lower with respect to normal values, which by itself provides strong additional evidence that low serum AMH amounts directly correlate with inefficient ovulation in women, since there was no history of pregnancies in all patients in our study. Once again, besides the multifactorial origin of POF in women, the low AMH concentration in serum is an independent marker of infertility in both young and adult women during their fertility periods.

Since the original ovarian cortex fragments were inadequate to isolate numbers of OSCs suitable for installing relative cultures, we cultivated those cells only in 16 instances. In relation to the size of the original cortex specimen, we obtained a variable absolute number of OSCs, but their percent occurrence was revealed using flow cytometric analysis that found an occurrence up to almost 15% of OSCs within the entire ovarian cell population. However, enrichment that was higher compared to their normal presence, which is usually about 5%, occurred only in two patients, thus supporting the hypothesis that the presence of OSCs in women with POF is independent from low serum levels of AMH, since, as expected, this hormone is ineffective on stem cells.

A similar variability in the number of OLCs formed was observed in vitro in OSC cultures. In particular, in several cultures, the formation of OLCs in single microchambers, namely the wells in the culture plate, ranged up to almost 30 OLCs, which represented an excellent result as compared to the original number of cultured OSCs from each patient. In fact, we calculated a mean number for each well of approximately 9.7 OLCs, which, if multiplied for the wells of the culture, provides a final number of OLCs of several tens for each patient, which is of course related to the original size of the processed ovarian cortex fragment.

Our data appear to support the results from similar studies by others. They definitely suggest that OSCs derived from the ovarian cortex are functional producers of OLCs in vitro and that these cells are obtainable at discrete quantities for female patients affected by POF or CTRI.

## 5. Future Application of OSCs for Female Fertility

The results from our study are in agreement with previous observations pursuing the suitability of OSCs in FP future applications, since these cells from POF-suffering females are capable, under appropriate culturing conditions, of generating haploid OLCs showing functional patterns as mature oocytes in their MII phase. At present, the number of further studies is growing, and the adoption of solid supports for OSC in vitro growth is being investigated. Within these, the utilization of MSCs in a feeder layer of the well-bed for OSC maintenance, as in our experiments, is considered functional, since these cells have been demonstrated to recreate the vascular bed in transplanted ovaries and in cultures with a solid support such as Matrigel.

Thus, it appears essential to pursue these studies, since the scientific technology, especially in the stemness field, is continually evolving. Today, the application of OLCs derived from OSCs in FP programs undoubtedly provides great opportunities and advantages, since in females with POF, the neo-oogenesis induced in vitro by OSCs does not imply unsafe procedures for patients, such as hormone burden to promote folliculogenesis for the eggs’ recruitment. This aspect is critical in women with CTRI, particularly those affected by hormone-sensitive cancers such as the majority of breast and gynecologic tumors, in which hormone treatments are probably able to accelerate tumor growth. In fact, oocyte cryopreservation is still proposed to patients as the most successful method to obtain pregnancies after healing from cancer, but the potential risk of stimulating the tumor via estrogenic charge is under-evaluated. Such a possible danger is frequently ignored when planning FP programs for females with cancers for whom drug ovarian suppression by GnRH agonists would probably be risk-free with respect to estrogenic stimulation. Furthermore, there is also a timely question that deals with the urgency for anticancer treatments required from specific patients. In fact, in females needing urgent neoadjuvant chemotherapies, the three weeks of time required to induce egg maturation after hormone burden and subsequent cryopreservation inevitably delays the oncologic treatment program. For females with cancer, ovarian cortex cryopreservation and reimplantation is probably a procedure free of oncogenic hormone-induced risks, but possible drawbacks primarily concern the eggs’ viability in the defrosted cortex rather than the potential occurrence of tumor cells within the cortex fragment to be reimplanted.

On the contrary, utilization of OLCs from OSCs would overcome these risks and restrictions in both POF and cancer patients. In fact, the laparoscopic biopsy of the ovarian cortex leads to isolation of OSCs with subsequent differentiation and in vitro expansion of large populations of OLCs, from which only those of high viability and quality will be selected for cryopreservation. Overall, this practice, which appears safe and risk-free in terms of tumor activation, allows us to select the most viable oocytes to be cryopreserved, since high numbers of oocytes are easily available for each patient.

However, further work is needed before translating the OSC technology to FP clinical application. In this regard, it is indeed necessary to verify whether or not the manipulation of OLCs promotes biologic alterations in maturing oocytes, and this requires additional studies exploring their genetic integrity before OSC technology translation to clinical use in FP programs. On the other hand, the efficiency of the OSC technology also needs to be accredited by the relative health authorities.

## 6. Conclusions

Infertility affects a large female population worldwide and is undoubtedly a disease. It is associated with a number of pathophysiological conditions such as POF, and its pathogenesis is sometimes undefined, whereas CTRI frequently recurs in young and adult females suffering from cancer. OSCs from women with ovulatory derangements have been demonstrated to generate, under proper conditions, in vitro OLCs that are apparently suitable for IVF. Current data are limited in terms of translating our results to clinical studies in POF and CTRI, but they do offer novel possibilities of translating this stemness methodology for FP programs for infertile women. Data from the cohort of patients investigated in the present study support this potential application in the reproductive medicine field, with the purpose of restoring the capacity to conceive in females otherwise unable to pursue their vocation of motherhood due to their disease.

## Figures and Tables

**Figure 1 biomedicines-12-01139-f001:**
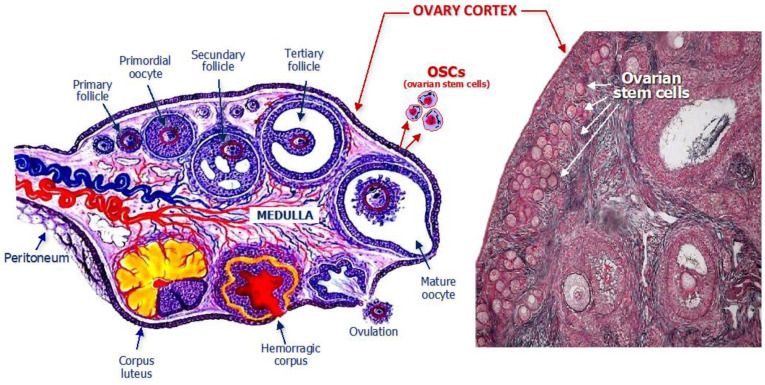
Structural anatomy of a woman’s ovary with emphasis on the location of ovarian stem cells which are normally residents within the cortex, as depicted in the schematic drawing (**left**) and detected in the histologic specimen (**right**).

**Figure 2 biomedicines-12-01139-f002:**
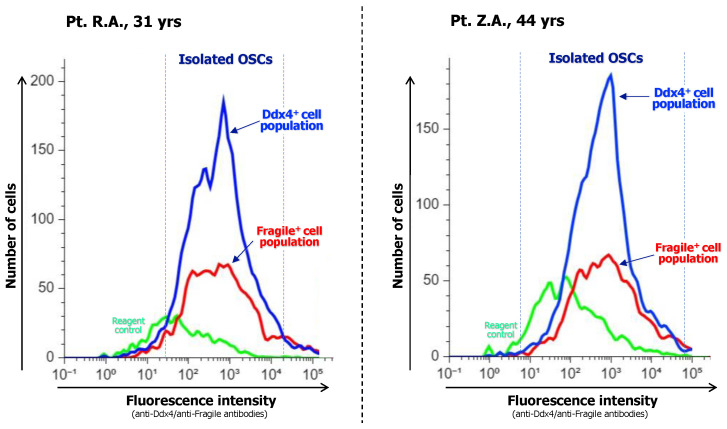
After immunoadsorption of OSC populations, their purity is evaluated with flow cytometry analysis measuring their enrichment in both Ddx4^+^ and Fragile^+^ subsets as specific OSC markers. Representative analyses from two patients are shown. Both cases depict the major content of isolated OSC fractions as the number of recruited cells and fluorescence intensity, with Ddx4^+^cells included at a higher magnitude with respect to the Fragile^+^ ones, thus suggesting that immunoselection was highly efficient in isolating the full OSC population.

**Figure 3 biomedicines-12-01139-f003:**
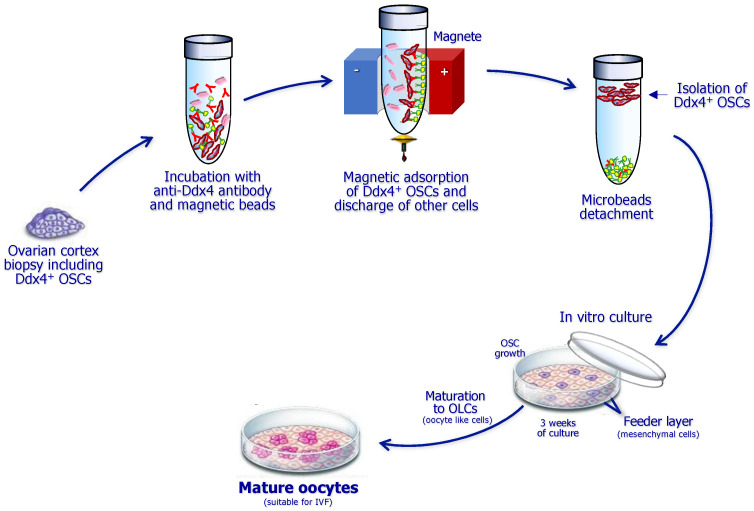
Sequential steps for isolation and culture of ovarian stem cells (OSCs) expressing the Ddx4 molecule. The ovarian cortex fragment is primarily lysed to obtain a cell suspension, which is then incubated with anti-Ddx4 antibodies conjugated to magnetic beads and further immunoadsorbed to discharge other cells. Thus, isolated OSCs are cultivated on a feeder layer including mesenchymal cells from the patient. After three weeks of culture with specific growth factors, they differentiate to large oocyte-like cells, namely, mature oocytes suitable for ICSI (intracytoplasmic sperm injection).

**Figure 4 biomedicines-12-01139-f004:**
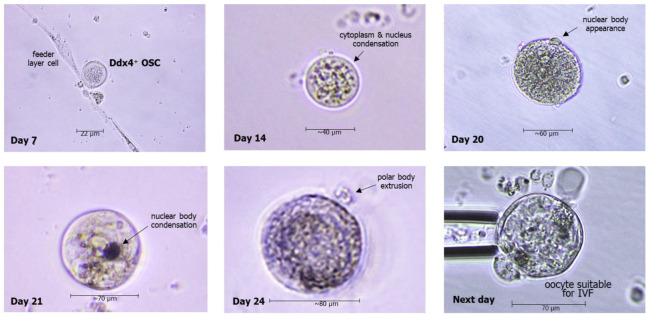
Representative microscope images of morphologic changes of OSCs in cultures, leading to generation of oocyte-like cells (OLCs) with extrusion of polar bodies after three weeks, thus entering the MII phase with the final shape of the mature oocyte.

**Table 1 biomedicines-12-01139-t001:** Population of females investigated, mean hormonal levels, and numbers of OLCs obtained from their OSC cultures.

	Number	Range	Mean
**Patients**	21	-	-
**Cause of infertility**	POF—20 pts ^1^ Oncology—1 pt ^1^	-	-
**Previous pregnancies**	None	-	-
**Age**	-	yrs ^2^ 30–44	36.5 ± 3.0
**AMH**	18/21 pts ^1^	0.01–0.56 ng/mL ^3^ (n.v: 0.1–1 ng/mL ^3^)	0.19 ± 0.03 ng/mL ^3^
**Ddx4^+^ OSCs collected**	16/21 pts ^1^	2.03–14,3%	7.43 ± 2.9%
**Installed OSC cultures**	16/21 pts ^1^	-	-
**OLCs obtained**	-	3–29 OLCs/mc ^4^	9.7 ± 1.3 OLCs/mc ^4^

^1^ pts: patients; ^2^ yrs: years; ^3^ ng/mL: nanograms/milliliter; ^4^ mc: microchamber.

## Data Availability

The data that support the findings of this study are available on reasonable request from the corresponding author due to continuing research and forthcoming publication of larger studies on this topic.

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
