# Peer review of "Ovarian Stem Cells for Women’s Infertility: State of the Art"

_biomedicines, 2024, doi:10.3390/biomedicines12061139_

Round 1

Reviewer 1 Report

Comments and Suggestions for Authors

The author reported their research findings on ovary stem cells, but many aspects were unclear in their manuscript.

The abstract does not adequately reflect the content of the study.

The abbreviations used in the article need to be consistent. For example, "ovary stem cell" appears as both OSC and OCS.

Is the intentionally highlighted mosaic on the left side of Figure 1? It's unclear what the author intends to convey.

Some parts of Figure 3 appear to be plagiarized, with similarities to Figure 3 of the article "Derivation, Characterization, and Differentiation of Human Embryonic Stem Cells," published in February 2004; Stem Cells 22(3):367-76. Please refrain from plagiarism. The author must confirm the true appearance of ovarian stem cells and provide authentic photos.

The author's statement in Figure 4, "Gene analysis of OLCs from previous studies have shown that OLCs express genes of mature oocytes and that, thus, should be suitable to be injected for fertilization with the male gamete," is purely speculative. Has there been any functional testing for fertilization?

Figure 4 is very blurry, making it difficult to discern. Please provide clear photos. The author should present more realistic and clear research results.

Comments on the Quality of English Language

 Minor editing of English language required

Author Response

Dear Reviewer,

Best regards,

Katarzyna Idzik 

Reviewer 2 Report

Comments and Suggestions for Authors

The authors present a manuscript which aims to investigate the role of ovarian stem cells in the preservation of female fertility by presenting their own data alongside a thoough review of literature. Although the manuscript is well written and offers interesting information, several corrections should be made to achieve better comprehension. First,  typographical errors throuhout the manuscript should be corrected (EU: European Union, Health Authorities: health authorities etc). Second, the first paragraph of the introduction part should be simplified and shortened to avoid confusion. Third, the authors should mention about the possible limitations of their findings. These limitations include the inclusion of only one patient with cancer related infertility (for authors' own data), lack of data about cost effectivity, risk of failure in the procurement of ovarain stem cells and lastly, the uncertainity about the success of achieving clinical pregnancy and live birth in future. I recommend that the revised version of this manuscript can be accepted for publication in Biomedicines.

Comments on the Quality of English Language

The authors present a manuscript which aims to investigate the role of ovarian stem cells in the preservation of female fertility by presenting their own data alongside a thoough review of literature. Although the manuscript is well written and offers interesting information, several corrections should be made to achieve better comprehension. First,  typographical errors throuhout the manuscript should be corrected (EU: European Union, Health Authorities: health authorities etc). Second, the first paragraph of the introduction part should be simplified and shortened to avoid confusion. Third, the authors should mention about the possible limitations of their findings. These limitations include the inclusion of only one patient with cancer related infertility (for authors' own data), lack of data about cost effectivity, risk of failure in the procurement of ovarain stem cells and lastly, the uncertainity about the success of achieving clinical pregnancy and live birth in future. I recommend that the revised version of this manuscript can be accepted for publication in Biomedicines.

Author Response

Dear Reviewer,

Yours faithfully,

Katarzyna Idzik 
